# Russian Companies' Motivations for Making Green Investments

**Liudmila S. Kabir *** and **Ivan D. Rakov**

International Finance Centre, Financial Research Institute of the Ministry of Finance of the Russian Federation, Nastasyinsky Lane, 3, b. 2, 127006 Moscow, Russia

*   Correspondence: lkabir@yandex.ru

**Abstract:** The purpose of this study is to identify the most significant motivations for Russian companies to make green investments. This article presents a multiple regression model based on panel data, designed to assess the impact of various factors on green investments made by Russian companies. To create this model, the authors used annual data for 83 regions of the Russian Federation for the period from 2011 to 2020. According to calculations made in this paper, the growth of green investments in the economy is due to the inflow of foreign direct investment, the increase in the collection of fees for negative impact on the environment, the increase in the production of extractive products and the growth of $CO_2$ emissions. At the same time, the total volume of investments is not affected by indicators assessing the environmental factor, but is affected by the inflow of foreign direct investments and the level of business concentration. The obtained results mean that the main motivators that encourage Russian companies to make green investments today are the opinion of foreign investors, global decisions to reduce greenhouse gases and the partial tightening of national environmental legislation. This indicates that the degree of a companies' integration into the global economy is of great importance for its propensity to make green investments in Russia. Therefore, special approaches are needed from the state in order to create incentives for green modernization of the national economy. This study expands our understanding of the role that green investments can play in the economy and the motivation for companies to make them, thus contributing to the existing literature on this subject.

**Keywords:** green innovations; green investments; sustainable development; development factors; companies' motivations

## 1. Introduction

*1.1. Setting the Problem and the Goal of the Study*

In this study, we will examine the motivations of Russian companies to make green investments. The problem is significantly influenced by the political factor, since the green economy is a process that has emerged within the concept of sustainable development. It is promoted from the global level through joint decisions and efforts of countries within the framework of UN agreements. There is a global plan of action for 2030 and its related goals,[1] and this plan has not been revised by countries so far.

The key point of the green economy is to ensure the transition of industrial production towards a new technological order, which will allow for avoiding the crises manifested today and observed in the past. The solution of the problem of transition to a green economy allows for solving the problem of modernization (Altenburg and Assmann 2017). This explains the special role of green investments—investments in new innovative technologies (innovations) providing green modernization.

The great role of investments is justified by the fact that they launch the mechanism of structural economic transformation, which is the basis for restarting the economic growth model (Mingaleva and Gataullina 2012; Reilly 2012). It turned out to be natural to pay close

attention to the question: what influences companies' decisions to make green investments, taking the country to a new level of development and prosperity?

Thus, we aim to determine what is the stronger motivation for Russian companies in making green investments. Is it government incentives or the company's own business strategy?

### 1.2. Research Gap

In this study, we examine whether Russian companies have strong incentives to make green investments and what these incentives are.

The analysis of the Russian-language segment of research presented on the eLibrary platform[2] leads to the conclusion that the financial instruments of green investment is a popular topic (Altunina and Alieva 2021; Bezsmertnaya 2021). Here, the research interest is formed around the study of foreign experience and comparison of the emerging Russian practice with this experience (Beloshitskiy 2021; Yakovlev and Nikulina 2019). Researchers have analyzed the dynamics of green stocks (Mikhaylova and Ivashkovskaya 2020); bonds and loans (Nikonorov et al. 2021; Smirnov 2021); government subsidies and green budget spending (Boltinova 2022); and rules regulating green finance markets (Chernikhovskaya 2022). Recently, another area of research on green financial instruments has emerged: green crypto assets (Baboshkin et al. 2022). The experience of individual companies in attracting green investments was discussed (mainly on the example of the transport, energy and waste processing industries) (Amirova et al. 2021; Kleandrov 2022; Mingaleva and Shpak 2015; Satsuk and Lobodina 2022). The ESG agenda has also been widely discussed as part of the green investments issue (Kormishkina et al. 2022; Kurnosova 2022; Popova and Strikh 2022; Tsygalov 2022), since it relates to the problem of securing funding sources for the company's green growth. Incentives for companies to increase their adoption of responsible environmental and green practices were explored. At the same time, the focus was on companies from resource-intensive industries and the problems of their technological re-equipment and innovation activities (Danilina and Mingaleva 2013; Tchaikovsky 2021).

There is a discussion about the measures of state support for green investments. Based on the analysis of foreign experience, researchers make proposals for increasing the role of the Russian state in this process, as well as for the formation of new institutions and the institutional environment of green investments (Mokhov and Chebotareva 2022).

Another topic of interest is the measurability of green investments. Here, scientists analyzed existing statistical indicators and drew conclusions about the applicability of some of them to measure and evaluate green investments. Often, this question refers to the evaluation of green investments on a national scale (Bobylev et al. 2015; Kuvalin et al. 2022).

The available research base forms a voluminous but fragmented picture (in the context of individual financial transactions, types of activities, regions of Russia) of the extent of spread of green economy principles. The focus is on foreign experience and the first Russian practices. At the same time, the motivations of Russian companies to make green investments and the factors explaining these motivations are not discussed in the current research.

In general, Russian researchers agree that the economic development of the country does not comply with the principles of sustainable development and green economy (Tagaeva et al. 2022). The consensus is that the situation can be changed through the intensification of green investments. At the same time, it has been noted that the existing administrative and economic tools of formation and regulation of such investment behavior are absent or do not function properly. This necessitates a detailed study of the factors affecting the investment process in order to understand the current motivations of Russian companies to make green investments.

### 1.3. The Value of the Research and Its Implications

The value of this study is determined by the problems it solves. Drawing on actual data of the Russian economy, we seek to identify the factors that influence green investments and

to answer the question: what is the most significant motivation for green investments by Russian companies? This will expand our understanding of the role that green investments play in the economy and the factors that motivate companies to make them. In addition, it will allow us to formulate proposals for the government to regulate processes in the field of green investments.

The main contribution of this study to the research field is the analysis of the factors affecting the green investments of companies, which makes it possible to provide an objective assessment of the current situation. The work carried out contributes to the elimination of empirical gaps observed in the study area, and allows for drawing conclusions about the relationship between the indicators used in the constructed model. The article also allows for the forming of an opinion on the role of administrative and economic instruments of formation and regulation of green investment behavior of Russian companies. In addition, this study expands the literature on the experience of different economies and different approaches to stimulating green investments. Overall, this article contributes to a broader field of research on the problem of economic growth and industrial development.

The following text is organized as follows: Section 2 presents the literature review and formulates the research hypotheses; Section 3 reveals the data used and describes the econometric research methods; Section 4 demonstrates the results of the calculations performed; Section 5 tests the robustness of the model; Section 6 presents the discussion; Section 7 provides the conclusions.

## 2. Research Concept and Literature Review

Developing a policy of the green modernization of the national economy requires not only knowledge of foreign experience, i.e., what the algorithms for launching the process of greening the economy are, its benefits and costs in the example of other countries, but also knowledge of how the principles of green economy have already penetrated the real sector of the Russian economy today. This is necessary in order to understand through which problem it is more expedient for the country to integrate into the international agenda and on which to concentrate the resources of business and government.

In this study, the term "green investment" is used, which refers to investments of companies in technological modernization. In this case, the priority is given to green technologies, which are innovative technologies in their essence. Therefore, the reference to the studies discussing the problem of green innovations of companies seems to be justified. Moreover, various researchers also tend to identify under green innovations a wide range of company activities that take into account environmental and green aspects (Tang et al. 2018).

We first describe the overall picture of green economy regulation in Russia after 2020, the year in which the country announced its ambitious plans in this area. Next, we review the literature, revealing current approaches to the study of factors that determine the propensity of businesses to make green investments.

### 2.1. National Green Investment Regulatory Policy

The implementation of green economy plans started with the adoption of the Presidential Decree No. 666 of 4 November 2020 "On reduction of greenhouse gas emissions". Both the public administration and the business community almost immediately came to understanding of the key issues that need to be addressed to achieve the goal of climate resilience and green development. Since 2020, regulation has been developing in this direction. We highlight a few areas there as follows.

First, these are quite rigidly discussed issues of budget expenditures, which are under the direct control of the Ministry of Finance of Russia:

1.   The requirement to amend the Tax Code of the Russian Federation: the application of a 0% rate from 2022 to 2024 to the coupon income on stable (including green) bonds placed during this period;

2. Issuance of targeted government green bonds to support infrastructure projects and innovative projects, including those in the field of renewable energy.

Second, these are measures related to the problem of economic development, which is supervised by the Ministry of Economic Development of Russia:

1. A mechanism for applying the criteria for sustainable (including green) development projects;
2. A system of verification of investment programs of industrial enterprises for the purpose of stimulating the reduction in carbon dioxide emissions;
3. Subsidies to Russian credit institutions and the state corporation VEB.RF for compensation of lost income on loans issued at a preferential rate to carry out measures to reduce emissions of pollutants into the atmosphere, as part of the implementation of the federal project "Clean Air" and the national project "Ecology";
4. Inclusion of R&D in the mechanism of granting subsidies to reimburse a portion of the costs of paying interest on loans implemented by enterprises that have a significant negative impact on the environment.

Third, these are measures related to the regulation of the financial market (Bank of Russia):

1. Development of measures to encourage financial market participants to conduct environmentally responsible financing;
2. Creation of a legal framework for the development of a market for corporate bonds with key performance indicators linked to sustainable development goals, as well as transitional climate bonds to finance projects of carbon-intensive companies aimed at improving their environmental friendliness;
3. Creation of adaptive financial instruments in a special segment of the Moscow Exchange.

Fourth, issues of accounting, reporting and auditing regulation:

1. Standards of information disclosure and nonfinancial reporting by participants in the market of green finance instruments;
2. The methodology of verification of compliance by issuers of green financial instruments with their obligations.

The fifth area is insurance. The unification of requirements for the formation of financial reserves by enterprises, intended for the elimination of accidents and emergencies, has become an urgent problem in environmental insurance.

These are the issues of national regulation that are being actively discussed today. Thus, the following hypothesis can be put forward:

**Hypothesis 1 (H1).** *Russian companies, when making green investments, are sensitive to government regulation.*

### 2.2. The Importance of Managers and Stakeholder Pressure

In foreign studies, the problem of motivating companies to invest in green innovations has various aspects. Much attention is paid to the role of company management in the formation of a green agenda and motivation for green innovations. It is noted that investments in green innovations have two important dimensions: an instrument of environmental protection and, in fact, innovations providing companies with competitive advantages and sustainable growth (Luo et al. 2021). However, the attractiveness of investing in green innovations is reduced by the fact that it is often perceived as an environmental cost (Rennings and Rammer 2011). Financing these costs by the company is beneficial to society, but for business it is an investment that has risks, pays off within a certain period and for which the head of the company is financially responsible (Lv et al. 2021). As a result, company management tends to avoid making decisions on investing in green innovations. To overcome this problem, it is necessary to find a motivation that will allow the head of the company to reduce his desire to avoid investment risks. Fairness, reputation and rights of control stand out as such incentives (Jayaraman and Milbourn 2015; Masulis and Mobbs

2014). Additionally, one possible solution is a contract with the head of the company that links the income of the manager to corporate growth based on effective innovative activity (Morck et al. 2005; Tauseef Hassan et al. 2021). In the same area of concern are studies that aim to determine whether there is a difference in incentives for leaders of private and public companies to pursue green innovations (Chen et al. 2011). The issue is of particular relevance for developing countries where the level of corruption is recognized as high, since the behavior of company managers is also influenced by such a factor as corruption (Wang et al. 2023).

Another direction of research is of the influence of managers' personal attitudes to the green agenda and the influence of stakeholders on the behavior of companies in the field of green investments (Wan et al. 2022). Researchers revealed the relationship of managerial consciousness and social responsibility of enterprise management with green investment decisions. Another force that influences managers' motivation to make green investment decisions is considered to be stakeholders (Jayaraman et al. 2023). They are increasingly interested in the green agenda and are able to have a serious impact on the economic performance of companies, forcing them to make certain strategic decisions. Since the company's strategy is determined by top managers' vision of green investments and stakeholders' concern about this issue, their influence should also be taken into account.

Thus, the motivation of the head of the company to make green investments is an important factor, and the pressure of interested parties (shareholders, investors, suppliers, customers) also fulfills its role.

### 2.3. The Global Agenda and Sustainable Development Goals

The impact of the global Sustainable Development Goals (SDG) agenda on companies' decisions to implement ambitious green investment plans is a new area of research that is being actively explored today. Here, attention should be paid to the publications that prove the greater involvement of large companies in the global agenda (Singh et al. 2020). This is also due to the fact that large businesses have environmental and social management systems, which encourages them to follow the goals of sustainable development and introduce green innovations. Seeing for themselves the benefits of activities focused on sustainable development, large businesses are motivated to implement green investment projects (Cuerva et al. 2014; Khattak 2020). The first studies are emerging that examine green investment practices associated with the fulfillment of company financial goals and commitments to sustainable development goals (Khan et al. 2022; Ullah et al. 2021). The conclusions of the research are that a proactive approach is needed to connect investment activities, achievement of the SDGs and the planned financial indicators, and that stakeholder pressure plays a major role here. The adoption of green investment reporting will eliminate tensions with stakeholders, increasing the company's accountability. At the same time, it will improve the company's achievement of both financial and SDG goals.

Thus, the incentives coming from the global level and from the external business environment are significant for companies when making green investment decisions. This allows us to formulate the following hypothesis:

**Hypothesis 2 (H2).** *The degree to which a company is integrated into the global economy is a significant factor influencing its green investment decisions.*

### 2.4. Environmental Regulation and Public Funding

Commitment to sustainable development goals is not enough even for large businesses. Both incentives coming from the global level and incentives coming from the state are needed, which can act in two ways.

First, there are government support measures, or government subsidies. They are especially important for companies in developing countries, as well as for small and medium-sized businesses, which are even more sensitive to government support (Ullah et al. 2021). The governments of countries, in turn, are also interested in both achieving

sustainable development goals and solving environmental and social problems. In this regard, the practice of implementing state programs, subsidizing, tax and other incentives linked to the achievement of national sustainable development goals is expanding (Albort-Morant et al. 2016; Azhgaliyeva et al. 2019; Eyraud et al. 2013; Owen et al. 2018; Rakov 2017; Tran et al. 2020). All these measures reduce the R&D costs of companies in green technologies, and also reduce the cost of green investments. Current studies (Li et al. 2022) make a strong argument that, in the absence of government support, green innovation leads to an increased economic burden on companies, restrictions on the applied technologies and a significant decrease in the economic efficiency of companies. To receive a positive effect from green innovations, governments should strengthen their support, implement the correct macroeconomic regulation and form a legal system of innovation protection. In turn, companies should focus on modernization of production processes and at the same time actively accumulate technologies in order to neutralize the negative impact of green innovations on economic efficiency.

The importance of public subsidy policy and information asymmetry for the effectiveness of green innovation stimulation is being actively studied (Liu et al. 2022). Thus, the analysis of state support has received considerable attention.

Second is the study of the role of environmental regulation as an incentive for businesses to make green investments. Here, we can find a wide range of studies that suggest to shift the focus from the study of traditional market behavior of companies to environmental regulation. According to research (Stucki et al. 2018), most enterprises do not see strategic prospects for initiating both environmental and green economy transformation. It has already been noted above that governments can stimulate companies by increasing their interest and willingness to make green investments. At the same time, it is widely believed that incentive policies must be backed by strict regulation to achieve the results for which those incentives are created.

The hypothesis formulated by Michael Porter in 1991 provides a theoretical framework for environmental regulation. It argues that stringent environmental regulations can encourage companies to invest in innovations that help improve commercial competitiveness. These purposes are served by environmental regulation widely used around the world (Berrone et al. 2013; Zafarullah and Huque 2018). As noted in a study (Ji et al. 2019), environmental regulation makes companies understand the importance of green investments and directs them to actively participate in them. In addition, the rising cost of pollution creates incentives for companies to transform and modernize their operations. However, the impact of environmental regulation is not so clear-cut. Although there are studies that confirm the positive impact of environmental regulation on green investment in certain industries (Giessen and Sahide 2017; Steinhorst and Matthies 2016), other researchers reveal the negative impact of environmental regulation on investment, which is manifested in higher business costs and reduced flexibility (Stucki et al. 2018). This limits the company's investment opportunities.

Perhaps the reason for the differences in the assessment of the role of environmental regulation is due to the varying degrees of its stringency. The stringency of environmental regulation has also been found to matter (Chen et al. 2022). Current research proves a U-shaped relationship between environmental regulation and investment in green technological modernization. This means that as the regulation tightens, its effect gradually changes from inhibition to stimulation (Song et al. 2020).

The impact of new market-based green economy instruments, namely carbon emission trading policy (CETP), is also the focus of current research. The key question is: can CETP influence green investment and how will this influence manifest itself? There is no single opinion in this area yet. There is also an assumption that the effects of this policy will differ from country to country. However, one of the recent studies (Wu et al. 2022) argues, using China as an example, that carbon trading policies encourage green investments by companies. This conclusion applies to nonstate companies, large companies and companies from clean sectors of the economy. At the same time, it is noted that internal policy

incentives, such as the cost compliance effect and the innovation compensation effect, have played the largest role in this process so far. That is, the very need to participate in the carbon trading system encourages companies to make green investments, while external factors, such as the price of carbon, liquidity and activity of the carbon market, do not serve as such incentives.

Recently, several studies have appeared that propose to evaluate the impact of the policies simultaneously stimulating and regulating green investments. For example, an article (Yu et al. 2022) describes the option when a system of subsidies for green innovations in certain industries is introduced along with a progressive carbon tax. The researchers argue that there is a synergistic effect of green innovation from those green innovation subsidies and carbon taxes. The benefits to the country's budget are lower subsidy costs, and the tax creates additional incentives for companies to adopt green innovations, while encouraging them to reduce carbon emissions. To increase the synergistic effect, the authors of the article proposed varying the carbon tax rate, rather than the share of subsidies.

Another study (Li 2022) examined the simultaneous impact of environmental subsidies and environmental taxes on the efficiency of corporate green investments. Having considered the mechanism of their joint influence on companies, the author concludes that it is necessary to improve the state supervision of green investments in order to increase the efficiency of spending public funds. At the same time, it is necessary to use the opportunities of the market mechanism to strengthen the supervision of environmental taxes and stimulate companies. It is important to create competitive conditions for companies to access state subsidies and to exclude rent-seeking activities by companies.

The analysis carried out in this study allows us to formulate the third research hypothesis:

**Hypothesis 3 (H3).** *Green investment flows are strongly influenced by public financing decisions.*

## 3. Evaluation Method and Data

### 3.1. Dependent Variable

There is no single approach to the statistical evaluation of green investments in the scientific literature. Siedschlag and Yan (2021) rely on the methodology described in the "Environmental Protection Expenditure Accounts Handbook: 2017 Edition". They propose to use the indicator "investment in fixed assets aimed at environmental protection and rational use of natural resources" (or environmental protection investments). This indicator is divided into two parts: (1) pollution control costs and (2) investments related to cleaner technologies. Chen and Feng (2019) and Khalid et al. (2023) used specialized indicators of corporate green investments calculated by national research centers. Tran et al. (2020) collected data on Vietnam's green investments through a survey of enterprises. Eyraud et al. (2013) offered their own calculation of the green investment indicator based on information accumulated in commercial databases. Shuai and Fan (2020) calculated a green economy efficiency index using the DEA model, which allows for evaluating the efficiency of consumed factors of production. The following indicators are used as input variables: employment in various regions, energy consumption and capital investments. Indicators of GDP and emissions of pollutants act as output data. Dutta et al. (2020) used the Environmental Protection Index (MSCI global environment index) and the green building construction index (MSCI global green building index), calculated by the international analytical company MSCI Inc., New York, NY, USA.

In this paper, the indicator of green investments is the investment of companies in fixed capital aimed at environmental protection and rational use of natural resources (environmental protection investments). This indicator has been available in the official statistical resources for each subject of the Russian Federation for a long period of time, which makes it preferable for our study compared to the other indicators listed above.

However, green investments do not only include investments in fixed assets aimed at environmental protection and rational use of natural resources. Any investment can have a positive influence here. For example, the installation of solar panels, the introduction of the

best available technologies, etc. Therefore, we additionally consider total investment in fixed capital as a dependent variable.

### 3.2. Independent Variable

The following are seen as the main factors influencing green investments: state policy in the subjects of the Russian Federation, the volume of products manufactured by enterprises, the export of Russian enterprises abroad, accumulated foreign direct investment in Russian enterprises, the state of the global situation, depreciation of fixed assets and emissions of pollutants.

It has already been discussed above that in various countries, government environmental policies and policy support measures are a key element in encouraging companies to make green investments. In this research, public policy is examined in two aspects: the level of environmental policy stringency (regulatory environmental measures, fines and taxes for negative environmental impact, etc.) and measures of state support for green financing (public spending on environmental protection, public green funding, etc.). The level of environmental policy strictness is assessed here using such an indicator as revenues received by the consolidated budget in the form of payments for negative environmental impact. In turn, the measures of state support for green financing are assessed through such an indicator as the amount of public spending on environmental protection.

Additionally, the econometric model includes the following indicators: the volume of pollutant emissions into the atmosphere and the discharge of polluted wastewater into surface water bodies. This is because the indicators presented above cannot cover all environmental regulation. At the same time, international environmental acts, international agreements and the growing role of corporate social responsibility, which motivate company management to invest in projects reducing pollutant emissions, play an equally important role in making investment decisions.

The processes of financial and trade integration of Russia into the world economy are no less important when considering the investment activities of companies in the subjects of the Russian Federation. Siedschlag and Yan (2021) have proven that large exporting firms that are part of a group of enterprises operating abroad are more likely to invest in environmental protection. Chai et al. (2021) proved that FDI has a positive impact on green growth in China's provinces. Therefore, such indicators as the volume of exports of goods and the volume of accumulated foreign direct investment in the subjects of the Russian Federation were added to the econometric model. To account the impact of business size and the presence of large firms in the region, the model includes such an indicator as the cost of fixed assets, taking into account depreciation.

The largest share of green investments is concentrated in the industrial production of goods, as industry is one of the sources of environmental pollution. Therefore, the model also contains an indicator of the volume of industrial production in the subjects of the Russian Federation in the context of the three main types of activity: mining; manufacturing industry; production and distribution of electricity, gas and water.

### 3.3. Regression Models

Identification of motivations of Russian business for green investments was carried out using a multiple regression model based on panel data. The regression equation was assessed using the least squares method and the estimated generalized least squares method (EGLS) for the random effects model. The selection of independent variables was carried out step by step, excluding the least significant variables or if there was a strong correlation between independent variables (multicollinearity). The regression equations were evaluated as a pooled model, fixed effects model and a random effects model. The validity of the estimates of the included individual effects were determined using a fixed effects test (Baltagi 2005) and the Hausman test for correlated random effects (Hausman 1978).

As a result, the following regression models were obtained for evaluation:

Model 1:

$$
\begin{aligned}
GreenInv_{it} = c + \beta_1 * Export_{it} + \beta_2 * FDI_{it} + \beta_3 * IndProdMining_{it} + \beta_4 \\
* IndProdManufacturing_{it} + \beta_5 * IndProdElectrAndWater_{it} + \beta_6 \\
* BudSpendEnvir_{it} + \beta_7 * PayNegEnvirImpact_{it} + \beta_8 * CO_{2_{it}} + \beta_9 \\
* Wastewater_{it} + \beta_{10} * FixedAssets_{it} + \varepsilon_{it}
\end{aligned}
\tag{1}
$$

Model 2:

$$
\begin{aligned}
Inv_{it} = c + \beta_1 * Export_{it} + \beta_2 * FDI_{it} + \beta_3 * IndProdMining_{it} + \beta_4 * IndProdManufacturing_{it} + \beta_5 \\
* IndProdElectrAndWater_{it} + \beta_6 * BudSpendEnvir_{it} + \beta_7 * PayNegEnvirImpact_{it} \\
+ \beta_8 * CO_{2_{it}} + \beta_9 * Wastewater_{it} + \beta_{10} * FixedAssets_{it} + \varepsilon_{it}
\end{aligned}
\tag{2}
$$

where *GreenInv* means investments in fixed assets aimed at environmental protection and rational use of natural resources, million rubles in 2015 prices;

*Inv*—investments in fixed capital, million rubles in 2015 prices;
*Export*—goods of own production that were shipped for export to foreign countries, million rubles in 2015 prices;
*FDI*—Accumulated Foreign Direct Investment in the subjects of the Russian Federation, million rubles in 2015 prices;
*IndProdMining*—industrial production volumes in the subjects of the Russian Federation by type of activity "Mining", million rubles in 2015 prices;
*IndProdManufacturing*—volumes of industrial production in the subjects of the Russian Federation by type of activity "Manufacturing", million rubles in 2015 prices;
*IndProdElectrAndWater*—volumes of industrial production in the subjects of the Russian Federation by type of activity "Production and distribution of electricity, gas and water", million rubles in 2015 prices;
*BudSpendEnvir*—expenditures of the consolidated budget of a subject of the Russian Federation for environmental protection, million rubles in 2015 prices;
*PayNegEnvirImpact*—accumulation of funds in the consolidated budget of a subject of the Russian Federation in the budget line "payment for negative environmental impact", million rubles in 2015 prices;
$CO_2$—emissions of pollutants into the atmospheric air from stationary sources, thousand tons;
*Wastewater*—discharge of polluted wastewater into surface water bodies, million cubic meters;
*FixedAssets*—the cost of fixed assets, taking into account depreciation, million rubles in 2015 prices;
$\varepsilon_{it}$—random error.

### 3.4. Data

Russia began to take the first steps towards greening the economy more than ten years ago. In 2010, the Decree of the Government of the Russian Federation No. 1016 "On Approval of the Rules for the Selection of Investment Projects and Principals for the Provision of State Guarantees of the Russian Federation for Credits or Bonded Loans Raised for the Implementation of Investment Projects" was adopted, which laid the foundation for the possibility of providing state guarantees to energy-saving projects. In 2012, the Fundamentals of State Policy in the Field of Environmental Development of the Russian Federation for the period up to 2030 were approved, where it was announced that environmentally oriented economic growth would be ensured.

Thus, the study of green investments in the Russian regions was carried out for the period from 2011 to 2020. Data for 2010, 2021 and 2022 were not taken due to the lack of their publication in official resources. The analysis was carried out for 83 subjects of the Russian Federation. Panel data were balanced. Preliminarily, all statistical data in monetary terms were adjusted for the consumer price index (2015 was chosen as the base year). Data on foreign direct investment and exports, presented in US dollars, were converted into

rubles. Data on the exchange rate and consumer price index were taken from the official website of the International Monetary Fund.[3]

We used information that is publicly available. Statistics on investments in fixed capital aimed at environmental protection and rational use of natural resources, on exports, as well as on accumulated foreign direct investment, were taken from the database posted on the Unified Interdepartmental Information and Statistical System (EMISS) website (https://www.fedstat.ru/, accessed on 15 December 2022). Data on the expenditures of the consolidated budget of a subject of the Russian Federation for environmental protection and on receipts to the consolidated budget of a subject of the Russian Federation in the line "payment for negative environmental impact" were taken from the database posted on the website of the Federal Treasury of the Russian Federation (http://datamarts.roskazna.ru/konstruktor/, accessed on 15 December 2022). The rest of the data were taken from the website of the Federal State Statistics Service (Rosstat) (https://rosstat.gov.ru/folder/210/document/47652, accessed on 15 December 2022).

It should be noted that data on accumulated foreign direct investment are presented on the EMISS website (https://www.fedstat.ru/indicator/31231, accessed on 15 December 2022) only for the period from 2011 to 2013. Therefore, it was decided to extend the time series from 2011 through information on incoming foreign direct investment in Russia (Rosstat statistics). At the same time, data on the outflow of FDI were not included in the calculations because they can distort the situation. The reason is that the outflow is dominated by direct investments sent by Russian residents abroad, rather than those withdrawn back by nonresident investors.

Detailed descriptive statistics for the dataset are presented in Table 1.

**Table 1.** Descriptive statistics.

|  | Observations | Mean | Median | Maximum | Minimum | Std. Dev. |
|---|---|---|---|---|---|---|
| GreenInv | 830 | 1784 | 550 | 23,371 | 0 | 2858 |
| Inv | 830 | 184,564 | 96,822 | 2,893,035 | 0 | 271,839 |
| Export | 830 | 140,865 | 35,246 | 3,228,247 | 0 | 316,404 |
| FDI | 830 | 93,295 | 17,086 | 2,912,993 | 0 | 298,473 |
| IndProdMining | 830 | 149,058 | 13,256 | 3,499,763 | 70 | 397,918 |
| IndProdManufacturing | 830 | 426,167 | 189,703 | 6,105,455 | 333 | 684,667 |
| IndProdElectrAndWater | 830 | 67,200 | 38,858 | 940,204 | 981 | 97,465 |
| BudSpendEnvir | 830 | 404 | 113 | 25,098 | 0 | 1470 |
| PayNegEnvirImpact | 830 | 238 | 112 | 3745 | −38 | 373 |
| $CO_2$ | 830 | 214 | 97 | 2583 | 0 | 377 |
| Wastewater | 830 | 170 | 84 | 1239 | 0 | 231 |
| FixedAssets | 830 | 1,146,128 | 547,875 | 35,823,308 | 23,376 | 2,663,108 |

The Levin et al. (2002) test for the stationarity of the panel data was performed beforehand on the variables included in the model. When testing, individual fixed effects were taken into account. The variables are found to be stationary except for IndProdElectrAndWater and FixedAssets. IndProdElectrAndWater and FixedAssets are converted to stationary variables by taking their successive differences D( . . . ) (Table 2).

**Table 2.** Unit root test (Levin, Lin and Chu).

|  | Statistical | Prob | Cross-Sections | obs |
|---|---|---|---|---|
| GreenInv | −50.5843 | 0.0000 | 83 | 726 |
| inv | −15.3367 | 0.0000 | 83 | 710 |
| Export | −8.00525 | 0.0000 | 81 | 702 |
| FDI | −9.15646 | 0.0000 | 83 | 744 |
| IndProdMining | −8.67141 | 0.0000 | 83 | 703 |
| IndProdManufacturing | −6.08783 | 0.0000 | 83 | 721 |
| IndProdElectrAndWater | 13.3875 | 1.0000 | 83 | 727 |
| BudSpendEnvir | −2.39080 | 0.0084 | 83 | 722 |
| PayNegEnvirImpact | −7.60805 | 0.0000 | 83 | 727 |
| $CO_2$ | −8.54869 | 0.0000 | 83 | 721 |
| Wastewater | −24.6791 | 0.0000 | 82 | 717 |
| FixedAssets | 6.32148 | 1.0000 | 83 | 699 |

Table 3 presents the pairwise correlation coefficients for the variables included in the model.

**Table 3.** Pair correlation coefficients.

|  | (1) | (2) | (3) | (4) | (5) | (6) | (7) | (8) | (9) | (10) | (11) | (12) |
|---|---|---|---|---|---|---|---|---|---|---|---|---|
| GreenInv (1) | 1.00 | 0.59 | 0.57 | 0.36 | 0.43 | 0.54 | −0.15 | 0.28 | 0.41 | 0.48 | 0.45 | 0.25 |
| Inv (2) |  | 1.00 | 0.84 | 0.77 | 0.58 | 0.83 | −0.27 | 0.63 | 0.42 | 0.31 | 0.63 | 0.58 |
| Export (3) |  |  | 1.00 | 0.83 | 0.53 | 0.83 | −0.18 | 0.58 | 0.26 | 0.27 | 0.51 | 0.47 |
| FDI (4) |  |  |  | 1.00 | 0.26 | 0.80 | −0.22 | 0.62 | 0.11 | 0.00 | 0.52 | 0.51 |
| IndProdMining (5) |  |  |  |  | 1.00 | 0.23 | −0.09 | 0.22 | 0.44 | 0.56 | 0.14 | 0.15 |
| IndProdManufacturing (6) |  |  |  |  |  | 1.00 | −0.24 | 0.65 | 0.29 | 0.18 | 0.75 | 0.53 |
| D(IndProdElectrAndWater) (7) |  |  |  |  |  |  | 1.00 | −0.18 | −0.07 | −0.07 | −0.15 | −0.31 |
| BudSpendEnvir (8) |  |  |  |  |  |  |  | 1.00 | 0.16 | 0.04 | 0.44 | 0.60 |
| PayNegEnvirImpact (9) |  |  |  |  |  |  |  |  | 1.00 | 0.68 | 0.52 | 0.01 |
| $CO_2$ (10) |  |  |  |  |  |  |  |  |  | 1.00 | 0.31 | 0.01 |
| Wastewater (11) |  |  |  |  |  |  |  |  |  |  | 1.00 | 0.32 |
| D(FixedAssets) (12) |  |  |  |  |  |  |  |  |  |  |  | 1.00 |

## 4. Results

The evaluation of the two models revealed that the volume of green investments in the subjects of the Russian Federation depends, all other things being equal, on direct foreign investments, production volumes of the extractive industry, fees charged for the negative impact on the environment and emissions of pollutants into the atmosphere. Additionally, investments in fixed capital in the subjects of the Russian Federation depend, other things being equal, on direct foreign investments and the value of fixed assets.

In addition, the estimation of the two regression equations showed the presence of individual fixed effects on the subjects of the Russian Federation. The Houseman test showed the inconsistency of estimates of models with random effects (Hausman 1978).

Next, the reliability of the obtained estimation results was checked. First, we checked for autocorrelation using the Wooldridge test (Wooldridge 2002). In particular, in the regression equation where the dependent variable is investment in fixed capital (Inv), autocorrelation was detected at the significance level of 1%. Therefore, in this regression equation, variables were taken for calculations in the form of consecutive differences D(…) in order to eliminate autocorrelation. Second, the obtained models were tested for heteroscedasticity (the Wald test) (Baum 2001) and cross-sectional independence (Pesaran's CD Test) (Pesaran 2004) (Table 4).

**Table 4.** Tests for heteroscedasticity, for autocorrelation in panel data, for cross-section dependence in residuals.

| Wooldridge Test for Autocorrelation in Panel Data (H0: No First-Order AUTOCORRELATION) | | |
|:---:|:---:|:---:|
| GreenInv | Inv | D(Inv) |
| F(1, 82) = 1.402 | F(1, 82) = 13.959 | F(1, 82) = 0.714 |
| Prob > F = 0.2399 | Prob > F = 0.0003 | Prob > F = 0.4005 |

| Modified Wald Test for Groupwise Heteroskedasticity in Fixed Effect Regression Model (H0: sigma(i)^2 = sigma^2 for All i) | |
|:---:|:---:|
| GreenInv | D(Inv) |
| $\chi 2$ (83) = $1.6 \times 10^8$ | $\chi 2$ (83) = $1.5 \times 10^6$ |
| Prob > $\chi 2$ = 0.0000 | Prob > $\chi 2$ = 0.0000 |

| Pesaran's CD Test (H0: No Cross-Section Dependence (Correlation) in Residuals) | |
|:---:|:---:|
| GreenInv | D(Inv) |
| $CD_p$ = 3.857 | $CD_p$ = 20.743 |
| Prob = 0.000 | Prob = 0.000 |

As can be seen from Table 4, the resulting models contain heteroscedasticity and cross-sectional correlation in the residuals. In this regard, the Driscoll–Kraay standard errors correction is applied, which is robust to spatial and temporal dependence (Driscoll and Kraay 1998).

Table 5 presents the results of the estimation.

**Table 5.** Assessment results (*** $p < 0.01$; ** $p < 0.05$; * $p < 0.10$).

|  | **GreenInv** | **D(Inv)** |
|:---|:---:|:---:|
| C | 403.526 (2.68) ** | 903.956 (0.26) |
| FDI | 0.003 (3.25) *** |  |
| IndProdMining | 0.003 (2.11) * |  |
| PayNegEnvirImpact | 0.722 (3.61) *** |  |
| $CO_2$ | 2.476 (6.37) *** |  |
| D(FDI) |  | −0.421 (−3.05) ** |
| D(FixedAssets) |  | 0.028 (2.26) ** |
| Cross-section fixed | yes | yes |
| Period fixed | no | no |
| Adj. R-sq. | 0.66 | 0.30 |
| F-statistic | 132,16 | 11.13 |
| Prob(F-statistic) | 0.000 | 0.005 |

Note: values in parenthesis are t-statistics; the result of regression with Driscoll–Kraay standard errors.

All obtained regression equations are significant according to Fisher's F-test at a significance level of 1%. The included explanatory variables and individual effects describe 66% and 30% of the total variation in green and fixed investment, respectively.

Thus, in the subjects of the Russian Federation, attraction of foreign direct investment, an increase in the volume of production of extractive industries and an increase in the collection of fees for a negative impact on the environment by 1 million rubles leads to an increase in green investments by 3 thousand rubles, 3 thousand rubles and 722 thousand rubles, respectively. An increase in emissions of pollutants into the atmosphere by 1 kiloton also leads to an increase in green investments by 2.476 million rubles. With regard to investment in fixed assets, the following factors stand out. The increase in the residual value of fixed assets by 1 million rubles ceteris paribus causes an increase in investments in fixed capital in the subjects of the Russian Federation by 28 thousand rubles, and the growth of foreign direct investment reduces investments in fixed capital by 421 thousand rubles.

## 5. Robustness Test

The robustness of the two models was tested by selecting 30 subjects of the Russian Federation with the highest and lowest average values for green investments and investments in fixed assets over 2011–2020. The analysis showed that the factors affecting the volume of green investments and the level of investment in fixed assets in the regions that are leaders in terms of their volume (investment active) and laggards (investment passive) are different, but the list of independent variables is almost identical to the overall estimate (Tables 5 and 6).

**Table 6.** Robustness test results (*** $p < 0.01$; ** $p < 0.05$; * $p < 0.10$).

| | GreenInv (Low-Level) | GreenInv (High-Level) | D(Inv) (Low-Level) | D(Inv) (High-Level) |
|---|---|---|---|---|
| C | 32 (0.38) | 1117 *** (4.21) | −1021 (−0.96) | 3726 (0.49) |
| FDI | | 0.003 *** (4.65) | | −0.454 ** (−2.92) |
| PayNegEnvirImpact | 0.701 * (2.20) | | | |
| $CO_2$ | | 5.474 *** (13.54) | | |
| IndProdManufacturing | 0.002 ** (2.55) | | | |
| D(FixedAssets) | | | | 0.027 ** (2.29) |
| D(IndProdManufacturing) | | | 0.241 * (2.06) | |
| Period fixed | yes | no | no | no |
| 2012 | −43.678 *** (−10.89) | | | |
| 2013 | −109.933 (−67.68) *** | | | |
| 2014 | −87.496 (−22.12) *** | | | |
| 2015 | −141.387 (−29.93) *** | | | |
| 2016 | −105.151 (−22.36) *** | | | |
| 2017 | −92.185 (−16.09) *** | | | |
| 2018 | −123.677 (−16.21) *** | | | |
| 2019 | −115.266 (−12.04) *** | | | |
| 2020 | −115.733 (−13.50) *** | | | |
| Cross-section fixed | yes | yes | yes | yes |
| Adj. R-sq. | 0.28 | 0.40 | 0.23 | 0.35 |
| F-statistic | 98.38 | 108.02 | 4.23 | 11.42 |
| Prob(F-statistic) | 0.000 | 0.000 | 0.074 | 0.005 |

Note: values in parenthesis are t-statistics; the result of regression with Driscoll–Kraay standard errors.

Thus, in the subjects of the Russian Federation with a low level of green investments, their volume depends on the fees charged for the negative impact on the environment and the volume of production of the manufacturing industry. In regions with a high level of green investments, their volume depends on the volume of foreign direct investment and the volume of emissions of pollutants into the atmosphere.

Investments in fixed assets in investment-active regions depend on foreign direct investment and the residual value of fixed assets. And in investment-passive regions, their volume depends on the output of manufactured goods.

Differences in the included explanatory variables for regions with different volumes of investment are explained by individual effects and high correlations between some indicators involved in the selection. For example, the exclusion of industrial production volumes in the subjects of the Russian Federation by type of activity "Mining" can be explained by the fact that only regions with a high level of investment are mining regions to much more extent. The IndProdManufacturing and FixedAssets indicators have a high level of pair correlation, which may indicate their interchangeability in a number of regions.

Despite the revealed differences between regions with different volumes of investment (level of investment activity), there are no significant contradictions between the models presented in Tables 5 and 6. This allows us to confirm the reliability of the original model (Table 5).

## 6. Discussion

The purpose of our study was to determine the motivations of Russian companies when making green investments. Based on an extensive review of international and domestic studies, the key factors stimulating the management of companies to implement a green investment policy were identified and three hypotheses were formulated (please refer to Section 2).

The model partially confirmed that government policy has an impact on the green investments of Russian companies (the first hypothesis). The influence of the level of strictness of environmental regulation was revealed. At the same time, the impact of public financing decisions on green investments (the third hypothesis) is not observed, since the relationship between budget spending on environmental protection and green investments has not been established. These findings confirm the results of the work (Liao and Shi 2018), which argued that tightening environmental regulations has a positive effect on green investments. The work (Rakov 2017) also proved that, in developed countries, green investments are influenced by the level of environmental policy strictness and public spending on environmental protection.

We see a directly proportional relationship between FDI and green investments. The dependence is observed in regions with a high level of green investments, where large export-oriented companies with raw material specifics are located. This confirms the results of the study (Siedschlag and Yan 2021), which proved that large exporting companies operating abroad are more likely to invest in environmental protection; this also confirms the conclusions about the positive impact of FDI on the development of the green economy made in the work (Chai et al. 2021). This indicates the validity of the second hypothesis.

Our model also indicates that the volume of green investments in investment-inactive regions depends on the volume of industrial production in the subjects of the Russian Federation by type of activity "Manufacturing". This conclusion is consistent with the findings of a study (Liao and Shi 2018) that noted the positive impact of China's regional GDP on green investment.

In addition, the model showed that the level of total investment in fixed capital does not depend on environmental factors. It also revealed the differences between investment-active and investment-passive regions in terms of factors affecting the level of green investment.

In particular, the following conclusions were made:

1. Large export-oriented companies in the raw material sector of the economy are predominantly located in regions with a large amount of green investments. Here, a direct positive relationship between foreign investments and green investment, as well as green investment and $CO_2$ emissions, was revealed. At the same time, the growth of fees for the negative impact on the environment has no great influence on green investments in these regions. It can be concluded that investment decisions in these regions depend on the decisions of foreign investors. Moreover, international environmental regulation of $CO_2$ emissions is of greater importance than Russian environmental regulation. On the other hand, this positive correlation between green investments and $CO_2$ emissions confirms that investment-intensive regions are the main sources of $CO_2$ emissions. Additionally, these regions are home to large companies that can pay more attention to green investments. Similar conclusions were drawn in a previous study (Tudor and Sova 2021), where countries were grouped by income level in a study of $CO_2$ emissions.
2. It should also be noted that large export-oriented companies are mostly mining companies. This fact explains why green investment is influenced by the factor of the volume of industrial production in the subjects of the Russian Federation by type of activity "Extraction of minerals".
3. In regions with low volumes of green investments, in contrast to investment-active regions, investment decision-making is more influenced by Russian environmental regulation, assessed through the factor of payment for negative environmental im-

pact. The link between budget spending on environmental protection and green investments has not been established.

4.  Environmental regulation, assessed through the indicator of payment for the negative impact on the environment, does not affect the volume of investments in fixed capital, but affects green investments. This means that only part of the investment flow generated by companies can be supported by this motivation, while it plays a role mainly in investment-passive regions where companies not related to the mining activity are concentrated, the share of foreign investors is smaller, and exports are lower.

As a result, a complex situation is observed. The great dependence of the national green agenda on the international level of governance makes the prospects for its further implementation dependent on the global conjuncture and global politics, rather than on national development issues and business strategies. In the event of serious changes in the global agenda, we should expect serious adjustments in the field of policy as well, which will also cause changes in the mechanisms of influence on the behavior of national companies.

## 7. Conclusions

Understanding the reasons that encourage companies to invest in technological modernization (green investments) ensures the achievement of planned results and the solution of accumulated development problems. Moreover, the knowledge of what the main incentive to make green investments is helps optimize the time and resources spent on public policy in this area.

### 7.1. Contribution to Theory

The motivation of companies for green investments is the main condition for the technological modernization of the economy. This study advances the discussion about the influence of factors in different types of economy. Other studies note that for companies from developed and developing countries, the direction of the factors may be different, which accordingly leads to a mismatch of motivations. Our study draws attention to the fact that, within the perimeter of developing economies, the influence of factors on the propensity of companies to green investment is not the same. For example, there are differences between the Chinese economy, which has a strong industrial sector, and the Russian economy, which is characterized by a high proportion of extractive industries.

### 7.2. Management Input

Foreign investment, international regulation of greenhouse gas emissions and state environmental regulation are the first factors influencing the propensity of Russian companies to make green investments. However, these are not the only factors included in our model. We have defined a set of indicators that can measure these factors and presented a model for their evaluation. By interpreting the results of the calculations, we obtained the opportunity to assess the current situation in the economy and make managerial decisions. Thus, our study expands the toolkit of analysis for the purpose of making managerial decisions.

### 7.3. Limitations and Suggestions for Future Research

The findings of this study have certain policy implications for the development of green investment in Russia and the search for incentives for national companies. First, it is necessary to further develop the institutional environment for green investment in order to be able to generate clear signals for companies at the state level. The research presented in this article shows that the current state environmental regulation is not a signal for companies to make green investments. It is necessary to conduct research on the impact of individual measures of state environmental policy and their combination on the investment behavior of companies.

Second, in order to stimulate green investments on an economy-wide scale, the boundary between external factors (foreign investors, international regulation of $CO_2$ emissions, the global agenda) and internal factors (national development goals) should be determined.

It is especially important to correctly define the degree and manner of influence of foreign investors and national economic policies. It is important to continue to leverage the effects of foreign investment, while expanding the domestic investment opportunities of companies. In this regard, it is relevant to study the investment behavior of foreign investors, Russian companies and national policies in order to identify potential areas of controversy that may negatively affect the propensity of companies to green investments.

Third, taking into consideration the decline in financing of Russian companies and the downward trend in fixed capital and green investment, it is necessary to promote the expansion of public green investment to stimulate proper green investments by private companies and achieve the ambitious goals announced in 2020. Through the active development of national policy in this area, it is essential to improve the existing institutional environment for investment, scientifically justify methods of state support, and create conditions for the expansion of market financing of green investments. At the same time, we need to form an appropriate strategy for modernizing the economy on a new technological basis, capable of ensuring the transition to a sustainable green model of economic growth, defining the areas of responsibility of both the state and companies.

**Author Contributions:** L.S.K. developed the idea, hypothesis and overall writing; conducted the literature review. I.D.R. developed the model, conducted data analysis and description of results. All authors have read and agreed to the published version of the manuscript.

**Funding:** This research received no external funding.

**Institutional Review Board Statement:** This study was not funded and there was hence no need for it.

**Informed Consent Statement:** This research does not involve any human subject.

**Data Availability Statement:** This study is based on open data.

**Conflicts of Interest:** The authors declare no conflict of interest.

## Notes

[1] A/RES/70/1. URL: https://documents-dds-ny.un.org/doc/UNDOC/GEN/N15/291/92/PDF/N1529192.pdf?OpenElement (accessed on 10 February 2023).

[2] https://elibrary.ru/ (accessed on 10 February 2023).

[3] https://data.imf.org/ (accessed on 15 December 2022).

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
