# Peer review of "Russian Companies’ Motivations for Making Green Investments"

_jrfm, doi:10.3390/jrfm16030145_

Round 1

Reviewer 1 Report

It was with pleasure that I had the opportunity to read the article entitled "Factors of green transformation of Russian economy. What is more important: state policy or global conjuncture?".

Unfortunately, the article has some shortcomings that in my understanding prevent publication without a thorough review. Firstly, it presents a very colloquial and not very academic writing. Secondly, I think that there should be a better alignment with perspectives of (digital) transformation that justify the arguments of the authors. Third, I believe it is important to present relevant business cases that support the results of the article. Finally, I recommend readjusting the conclusions by organizing them into subsections - contributions to theory, managerial contributions, limitations and suggestions for future research.

Reviewer 2 Report

The manuscript deals with an interesting yet under-investigated topic. As such, it does deserve further attention. However, at this point, some revisions are needed to bring the manuscript to publication status.

Some of the elements that need to be revised are:

- the overall structure of the manuscript needs to be improved, as it is chaotic and hinders readability;

- the language also needs major improvement. Affirmations such as: "The paper under consideration...", "In the given paper,...", "In the given study..." should be eliminated

-the title and abstract need revision to better reflect the main goals and findings. In particular, the title has a faulty structure and should be simplified. The abstract should give more details on the estimators employed in the empirical investigation and better reflect the main findings, which are the most important part of a research paper.

- the introduction should better set the background for the research; it is important to identify gaps in the literature and show how the manuscript helps to fill them in; this would also help to highlight the contributions it makes to the extant literature. The current so-called contributions are not, in fact, contributions, but rather research findings. Overall, the introductory part needs major improvement.

-The third part of the study is chaotic; it should include the data and method; instead, a consistent literature review is also included and hinders its readability; it needs improvement to make it more readable.

--the data needs to be more presented, including the dimensions of the panel, information about its balancedness or lack of, etc.

- The method should be clarified. Appropriate preliminary testing should be performed and the best suitable estimators applied. Stationarity testing is needed for OLS/FE/RE estimators. This would inform on the need to transform some of the variables. Overall, this issue can affect the reliability of results. So, you cannot draw reliable conclusions following faulty estimations.

- The discussion of the results part is lacking; This is quite important, offering a reflection of similarities and differences with previous research, as well as the implications of the current findings and clarification of the research contribution. 

-conclusions should be more concise.

-the study should be based on more references. 

Reviewer 3 Report

The topic is timely

Some considerations must be met to make the article more understandable:

- The abstract must be improved. The reader does not know exactly the concrete objectives to be achieved and what gap it came to cover and the contributions and added value;

- The literature and hypotheses review section is poor and more recent articles should be added on the importance of the topic and its relationship with each hypothesis;

- The hypotheses should not be placed all in a row, but rather after the theoretical contributions for each one of them. That is, to reinforce the literature review for each hypothesis and the hypothesis to be placed only after these theoretical arguments. This will also facilitate the reading and discussion of the results,

- Following the previous point, the authors must make an effort to reinforce the discussion of the results in light of the literature review and with a macroeconomic framework.

- The arguments for the sample and period considered, as well as the variables to be included in the green investment model should have more support. For example, why do exports determine this type of investment?

- There should be a subsection for discussion of the results and it is not in the conclusions that the results are read. Conclusions should be more incisive and contain the limitations and possibilities for further work. The specific contributions to each type of stakeholder, manager, invested potential, regulator, civil society, etc... should also be reinforced in the conclusions.

Reviewer 4 Report

The paper draws attention to a very interesting research field and subject. Its relevance to the Factors of green transformation of Russian economy makes it even more interesting. Yet, the paper requires major changes as follows:

In the abstract session, I would suggest to better highlight the aim of this research together with its significance

In the introduction section, the author(s) should break down this section into a number of sub-sections. i.e., Research gap, Research aim, Value, and significance

Research gap: In general, a good discussion is noted to present the need for this study. However, there is room to further underpin and support your arguments. Supporting the core arguments related to the research gap based on more recent literature is instrumental to convey the message further related to the core value of your study.

Considering the nascent field of “green economies”, the range of extant literature covered is apt and wide. I would welcome just a few more contemporary works, such as:

Sustainable Entrepreneurship and Marketing Strategy: Exploring the Consumer “Attitude–Behavioural-Intention” Gap in the Sport Sponsorship Context

Empowerment and performance in SMEs: Examining the effect of employees' ethical values and emotional intelligence

Strategic Sport Sponsorship Management - A Scale Development and Validation”. Journal of Business Research

The chosen approach, instrument, and data analysis are well presented

Give more information regarding the choice of the methodology

I thank the author(s) for the opportunity to read this interesting article and I hope that the above recommendations shall assist them in improving  it.

Round 2

Reviewer 1 Report

Although the authors did not fully respond to my comments, I am satisfied with the authors' responses, particularly as the revision of the article was done with the due depth and taken seriously.

Reviewer 2 Report

I understand that efforts to improve the manuscript have been made. However, the previous recommendations have not been implemented in a satisfying manner.

The introduction is very long and segmented. I did underline the need to include some elements, including highlighting gaps in the literature, etc., but I did not mean over-segmentation of the text. Nonetheless, the gaps in the literature that the manuscript attempts to fill are not presented in a satisfactory manner. Other aspects need consideration, too. Very importantly, the method really needs clarification following a battery of preliminary testing. Also, there are common issues within the fixed effects panel models that cannot be overlooked, namely heteroscedasticity, autocorrelation, and cross-sectional dependence. These need to be tested and corrected. If detected, robust standard error computational techniques, such as the fixed effects estimator with Driscoll–Kraay standard errors, can be employed. If the problems still persist, then the estimator is faulty and the results are unreliable. 

Finally, the manuscript needs editing services. The language does not meet publication standards. 

Reviewer 3 Report

Since the authors accepted the suggestions and improved the article, I am of the opinion that it meets the conditions to be published.

Round 3

Reviewer 2 Report

The manuscript has been further improved and the methodological weaknesses have been mitigated. The novel dataset increases its originality and the readers' interest. Thus, I think it can be published after some more language polishing.
